# Use of Electrical Impedance Tomography (EIT) to Estimate Tidal Volume in Anaesthetized Horses Undergoing Elective Surgery

**DOI:** 10.3390/ani11051350

**Published:** 2021-05-10

**Authors:** Benedetta Crivellari, Anthea Raisis, Giselle Hosgood, Andreas D. Waldmann, David Murphy, Martina Mosing

**Affiliations:** 1School of Veterinary Medicine, Murdoch University, Murdoch 6150, Australia; benedetta.crivellari@gmail.com (B.C.); a.raisis@murdoch.edu.au (A.R.); g.hosgood@murdoch.edu.au (G.H.); d.murphy@murdoch.edu.au (D.M.); 2Department of Anesthesiology and Intensive Care Medicine, Rostock University Medical Center, 39071 Rostock, Germany; awa091086@gmail.com

**Keywords:** airway dead space, alveolar ventilation, equine, spirometry, volumetric capnography

## Abstract

**Simple Summary:**

The aim of this study was to explore the usefulness of electrical impedance tomography (EIT), a novel monitoring tool measuring impedance change, to estimate tidal volume (volume of gas in litres moved in and out the airways and lungs with each breath) in anaesthetised horses. The results of this study, performed in clinical cases, demonstrated that there was a positive linear relationship between tidal volume measurements obtained with spirometry and impedance changes measured by EIT within each subject and this individual relationship could be used to estimate tidal volume that showed acceptable agreement with a measured tidal volume in each horse. Thus, EIT can be used to observe changes in tidal volume by the means of impedance changes. However, absolute measurement of tidal volume is only possible after establishment of the individual relationship.

**Abstract:**

This study explores the application of electric impedance tomography (EIT) to estimate tidal volume (VT) by measuring impedance change per breath (∆Z_breath_). Seventeen healthy horses were anaesthetised and mechanically ventilated for elective procedures requiring dorsal recumbency. Spirometric VT (VT_SPIRO_) and ∆Z_breath_ were recorded periodically; up to six times throughout anaesthesia. Part 1 assessed these variables at incremental delivered VT of 10, 12 and 15 mL/kg. Part 2 estimated VT (VT_EIT_) in litres from ∆Z_breath_ at three additional measurement points using a line of best fit obtained from Part 1. During part 2, VT was adjusted to maintain end-tidal carbon dioxide between 45–55 mmHg. Linear regression determined the correlation between VT_SPIRO_ and ∆Z_breath_ (part 1). Estimated VT_EIT_ was assessed for agreement with measured VT_SPIRO_ using Bland Altman analysis (part 2). Marked variability in slope and intercepts was observed across horses. Strong positive correlation between ∆Z_breath_ and VT_SPIRO_ was found in each horse (R^2^ 0.9–0.99). The agreement between VT_EIT_ and VT_SPIRO_ was good with bias (LOA) of 0.26 (−0.36–0.88) L. These results suggest that, in anaesthetised horses, EIT can be used to monitor and estimate VT after establishing the individual relationship between these variables.

## 1. Introduction

Monitoring ventilation is essential to ensure adequate gas exchange during general anaesthesia. Ventilation is defined as the process through which a volume of gas is transported to and from the lungs. The gas volume that it is inhaled during a normal tidal breath is called tidal volume (VT). This volume can be divided into a part that takes part in gas exchange (alveolar tidal volume, VT_alv_) and one that does not (dead space). Dead space volume includes airway dead space (VD_aw_; volume in conducting airway) and alveolar dead space (volume in ventilated but not perfused alveoli).

Changes in VT_alv_ correspond to changes in alveolar (P_A_CO_2_) and arterial partial pressure of carbon dioxide (P_a_CO_2_), a variable than can be assessed via blood gas analysis and represents the gold standard of ventilation monitoring. However, blood gas analysis is an intermittent way to monitor ventilation. In clinical settings, ventilation adequacy on a breath-by-breath basis is assessed using end tidal carbon dioxide (P_ET_CO_2_). End-tidal CO_2_ values (in kPa or mmHg) are obtained non-invasively via capnography and are used to monitor adequacy of ventilation. In anaesthetised horses, however, ventilation/perfusion (V/Q) mismatch develops rapidly after induction [1]. Any increase in V/Q mismatch causes a broadened partial pressure gradient between P_ET_CO_2_ and P_a_CO_2_. Therefore, in anaesthetised horses, P_ET_CO_2_ is not a reliable indicator of ventilation adequacy [2,3].

Spirometry assesses gas flows, airway pressures and tidal volumes and allows continuous monitoring of respiratory mechanics [4]. For equine application, only few spirometers have been described. Some examples include the Fleish pneumotachograph, the ultrasonic-spirometry and the thermal mass flowmeter [5,6,7]. A piston driven large animal ventilator also measures tidal volume but has been shown to underestimate it by up to 20% [8]. To accommodate for the large flows and volumes encountered during large animal anaesthesia, human spirometry equipment has been adapted and/or remodelled for equine application [9,10,11,12,13]. For instance, a flow partitioning device can be used to split the flow in four smaller parts enabling measurement of larger volumes [9,10]. To date continuous spirometry monitoring is not yet consistently applied in equine clinical anaesthesia.

Electrical impedance tomography is a non-invasive thoracic imaging technology which measures the change in electrical potentials resulting from weak alternating currents applied at the surface of the body using an electrode belt [14]. The potential difference fluctuation between pairs of electrodes enables measurement of the change in electric impedance (∆Z_breath_) that occurs as air moves in and out of lungs over respiratory cycles [15,16].

The relationship between ∆Z_breath_, measured with EIT, and VT, measured with spirometry, has been investigated under standardised experimental conditions in a variety of species including dogs, pigs, humans and horses [17,18,19,20]. A strong relationship was found in dogs and pigs when incremental volumes were delivered [17,18] and this relationship persisted regardless of body position and anthropometric features in a human study [19]. Recently, EIT was applied in anaesthetised and mechanically ventilated horses to compare ∆Z and VT_SPIRO_ under experimental conditions, using a wide range of delivered tidal volumes between 4 L (8 mL/kg) and 10 L (20 mL/kg) [20]. This equine study also described a strong linear relationship between the two variables. Based on these finding, it was suggested that EIT may offer the potential to estimate tidal volume based on increases and decreases in impedance [20].

It is important to highlight that spirometry and EIT operate at different levels (Figure 1). Spirometric measurements are taken proximally, at the level of the incisors, and comprise the volume in the endotracheal tube and conducting airways where gas exchange does not occur (equipment and VD_aw_). In contrast, EIT represents a cross-sectional area of the lungs taken at the level of the 5–6th intercostal space and is unaffected by equipment and VD_aw_. Dead space volumes can be derived from volumetric capnography (VCap). Volumetric capnography plots the expired CO_2_ concentration over the exhaled volume of a single breath and uses Fowler’s method to calculate VD_aw_ and VT_alv_ [21,22]. For large animal applications, volumetric capnography and spirometry measurements are made possible by a flow partitioning device (FPD) [2,8,10].

The advantages of EIT over spirometry are that it provides continuous breath-by-breath measurements without the use of a face mask or intubation of the patient. EIT also avoids the drawbacks of spirometry, for equine applications, by accommodating large tidal volumes without the need to adapt existing equipment and not increasing equipment dead space. 

However, as it is not yet an established technology in veterinary medicine, EIT is a highly expensive monitoring tool for routine clinical anaesthesia.

The first aim of this study was to confirm the findings of the experimental equine study and determine the relationship ∆Z_breath_ and VT measured by spirometry (VT_SPIRO_) in horses anaesthetised for clinical purposes, over progressive increases in clinically relevant delivered tidal volumes (Part 1). It was hypothesized that a positive linear relationship between ∆Z_breath_ and VT_SPIRO_ would be present in all horses. 

The second aim was to use this relationship to estimate tidal volume (VT_EIT_) in litres from measured ∆Z_breath_ and compare the estimated VT_EIT_ with the actual measured VT_SPIRO_ (Part 2). It was hypothesised that VT_EIT_ would have good agreement with VT_SPIRO_ with limits of agreement within 20% of the measured VT [23,24]. 

To the authors’ knowledge, this is the first study in which the relationship between ∆Z_breath_ and VT_SPIRO_ was tested over a clinically relevant range of tidal volumes using a stabilization period between changes in the delivered volume. It is also the first study to estimate VT in litres from ∆Z measurements.

## 2. Materials and Methods

This study was approved by the Animal Ethics Committee of Murdoch University (permit number R3023/18) and performed according to the Animal Welfare Act of Western Australia (2003). All animals were cared for according the Australian code for care and use of animals for scientific purposes (2013). In addition, informed owner consent was obtained.

### 2.1. Animals

A total of 26 healthy, adult horses (ASA I and II), with no evidence of pulmonary disease, presenting for elective procedures requiring dorsal recumbency, were enrolled in the study. Horses were defined as healthy based on medical history, physical examination and routine pre-anaesthetic haematologic tests. Horses with ASA physical status greater than II, with abnormal clinic–pathologic findings or physical evidence of pulmonary disease, were excluded from the study. Young or unhandled horses, or those with a difficult temperament were considered unsuitable to wear an EIT electrode belt and excluded from the study.

### 2.2. Anaesthesia

Horses were fasted for at least 12 h before induction of anaesthesia but had free access to water until premedication. A 14-gauge catheter was placed in the jugular vein, on the morning of the procedure. The anaesthetic protocol (premedication and induction) was tailored to meet the specific needs of each animal and was left to the discretion of the anaesthetist in charge (Appendix A). Orotracheal intubation with cuffed silicone endotracheal tubes (ETT) was performed in all horses after induction. The horses were then hoisted into the surgical theatre and positioned in dorsal recumbency on the padded operating table.

The ETT was connected to a large animal ventilator (Tafonius Junior; Vetronic, UK) and volume-controlled mechanical ventilation (CMV) started.

Anaesthesia was maintained with isoflurane (IsoFlo, Zoetis Australia Pty Ltd.) delivered in oxygen. End-tidal isoflurane (ET_iso_) concentration was adjusted based on anaesthetic depth. Anaesthetic depth was assessed in relation to eye position, presence or absence of palpebral reflex, nystagmus and/or tear production. Routine monitoring was performed using a multi-parameter monitor (Solomon system, Vetronic, UK), and included electrocardiography, invasive blood pressure, pulse-oximetry, end-tidal carbon dioxide partial pressure, and concentration of inhaled and exhaled isoflurane. Physiological variables, expired and inspired agent % and ventilatory setting were recorded manually every 5 min. A catheter was inserted in the facial artery for invasive monitoring of blood pressure and blood gas analysis. Hartmann’s solution (Baxter Healthcare, Australia) was administered throughout the general anaesthesia at the rate of 5–10 mL/kg/hour. If mean arterial pressure (MAP) decreased below 70 mmHg, dobutamine (Hospira Pty Limited, Australia) was infused at 0.5–1.5 µg/kg/min IV and titrated to maintain MAP between 70 and 80 mmHg. 

### 2.3. Instrumentation

The chest circumference of each horse, at the level of the 5th and 6th intercostal space, was measured and recorded. Body condition score (BCS) was assessed from 1 to 9 as described by Henneke et al. [25]. After premedication, nonconductive ultrasound gel was applied on the unclipped skin around the thorax between the 5–6th intercostal space, directly behind the scapula. The custom made EIT belt was placed, under slight tension, around the thorax on top of the gel [26]. An elastic bandage was placed over the EIT belt to maintain its position and ensure all electrodes were in firm contact with the skin during induction and hoisting. The position of the belt and contact of the electrodes were visually inspected and corrected before the horse was positioned on the table. Once the horse was positioned on the surgical table, CMV was started. The start of CMV represented the beginning of the study. Initial ventilatory settings were set as follows: VT = 10 mL/kg body weight; respiratory rate (*f*R) = 8 breaths/min; inspiratory:expiratory time (I:E) = 1:2; positive end-expiratory pressure (PEEP) = 0 cmH_2_O. The belt was then connected to the EIT device and data recorded using a dedicated software package (BBVet, SenTec, Landquart, Switzerland). 

The accuracy of the used human spirometer (NICO, Respironics Inc. Murrysville, Pennsylvania, PA, USA) was verified with a calibration syringe. For the verification, volumes ranging from 100 mL to 500 mL were delivered in stepwise 100 mL increments. To enable the application of standard human spirometry to large animal anaesthesia, a Flow Partitioning Device (FPD) was used. The FPD was inserted between the Y-piece of the breathing system and the ETT. As the name suggests, this device splits the flow into four parts and directs it through four spirometry connectors, one of which is coupled with the spirometry monitor whereas the other were sealed to prevent leakage [9,10]. In order to correct for the flow splitting from the FPD, all volume measurements were multiplied by 4. The final set-up is represented in Figure 1.

### 2.4. Data Collection

Prior to data collection, leak volume was assessed and corrected if required. For this purpose, the difference between inspiratory and expiratory tidal volume was calculated. Additionally, the spirometry flow-volume curves were inspected. A difference of ≥10% identified by either the difference between the two tidal volumes or visual inspection of the flow-volume loop, was predefined as an unacceptable leak, and the cuff of the endotracheal tube was re-inflated. The final leak volume after correction was recorded for retrospective evaluation.

The study period comprised of a minimum of 4 to a maximum of 6 measurement points. The number of measurement points collected depended on the duration of the procedure.

*Part 1 (∆Z_breath_ and V**T**_SPIRO_**relationship)*:

The first part of the study investigated the relationship between change in impedance, measured with EIT (∆Z_breath_), and tidal volume, measured with spirometry (VT_SPIRO_). It comprised 3 measurements at tidal volumes of 10 mL/kg (M_10_), 12 mL/kg (M_12_) and 15 mL/kg (M_15_). A stabilisation period of 15 min was allowed in between measurement points, followed by a 2-min interval dedicated to data collection.

*Part 2 (Estimation of V**T**_EIT_**in litres)*: 

Based on the relationship determined in part 1 between ∆Z_breath_ and VT_SPIRO_, data collected at the subsequent measurement points were used to estimate VT (VT_EIT_). For this part, the delivered tidal volume was adjusted to maintain end tidal carbon dioxide tension (P_ET_CO_2_) at 50 ± 5 mmHg. Depending on the duration of the procedure up to 3 measurement points (M_A_, M_B_ and M_C_) were collected at 15 min intervals, followed by a 2-min window dedicated to data collection. Data collection ceased at the end of the surgical procedure or after M_C_. 

At each measurement point, EIT raw data were electronically recorded continuously over 2 min (16 breaths) while data for VT_SPIRO_ were manually recorded every 20 s (6 breathes). Volumetric capnography (VCap) data, alveolar ventilation (VT_alv_) and airway dead space (VD_aw_) were recorded once per measurement point [27]. 

Arterial blood gases samples were collected at M_12_ and M_A_ and analysed immediately after collection with the in-house blood gas analyser (ABL 800 flex, Radiometer medical Aps, Denmark). 

### 2.5. Data Extraction

For data analysis, a maximum of 10 stable and consecutive breaths were selected from the 16 breaths collected over the two minutes of EIT recording at each time point using a dedicated software (ibeX 2018, SenTec). 

Impedance change data (∆Z_breath_) was extracted from these breaths as well as variables to ascertain which factors may alter the relationship between ∆Z_breath_ and VT_SPIRO_. These included centre of ventilation (CoV_VD_) and dependent silent spaces (DSSs). 

Total impedance change was calculated for each breath (∆Z_breath_) by subtracting the impedance (Z) at the beginning of inspiration from the impedance at the end of inspiration. For each time point, the average ∆Z_breath_ from all analysed breaths was calculated. 

Centre of Ventilation (CoV) represents the focal area of the functional EIT image where ventilation occurs and is a parameter used to quantify the overall distribution of ventilation. The ventro-dorsal CoV (CoV_VD_) is generally expressed as a percentage of the ventro-dorsal extension of the predefined region of interest (ROI) within the functional EIT image that represents the lung regions [14,28,29]. The measured % varies between 0 and 100% where 0% represents ventilation being distributed within the most ventral aspect of the lung and 100% represents ventilation being distributed within the most dorsal lung region. For all horses, CoV_VD_ was calculated at each measurement point as an average of all breaths selected. 

Silent spaces refer to pixels within the ROI with an impedance change <10% of the maximal impedance change of any pixel within the ROI [30,31]. They are expressed as a percentage of the total lung region of interest. For the purpose of this study, only dependent silent spaces were calculated (DSS_S_). Dependent silent spaces comprise poorly ventilated (pixel in the ROI with 0 < ∆Z ≤ 10% of maximal ∆Z within ROI) or collapsed (pixel in the ROI with ∆Z = 0%) regions of the dependent lung. For all horses, DSS_S_ were calculated as an average of all breaths selected, at each measurement point.

Expiratory tidal volume (VT_SPIRO_) was recorded at each measurement point. Peak inspiratory airway pressure (PIP) was also recorded to determine its effect on the relationship between ∆Z_breath_ and VT_SPIRO_. Tidal volume measurements from the six representative breaths were multiplied by four to account for the split then averaged for each time point. 

Volumetric capnography variables which may have influenced the relationship were recorded. These included alveolar ventilation (VT_alv_) and airway dead space (VD_aw_) (mL) in 12 horses. Measurements were obtained using the VCap software integrated in the spirometer, which is based on Fowler’s method. As an FPD was used in this study, all VCap variables recorded were normalised with the recorded VT_SPIRO_ (VT_alv_/VT_SPIRO_; VD_aw_/VT_SPIRO_).

F-shunt, an oxygen index, was retrospectively calculated from arterial blood gas results and haemoglobin measurements at M_12_ and M_A_ to estimate the amount of venous admixture. F-shunt equation was calculated as follow [32,33,34,35]:(1)$$F-\mathrm{Shunt}=\frac{{{\mathrm{Cc}}^{\prime }\;O}_{2}-{\mathrm{CaO}}_{2}}{\left[{{\mathrm{Cc}}^{\prime }\;O}_{2}-\left({\mathrm{CaO}}_{2}-3.5\right)\right]}$$

Further details on the calculation can be found in Appendix B.

### 2.6. Statistical Analysis

Continuous variables were assessed for normality using a Shapiro–Wilk test with the null hypothesis of normality rejected at *p* < 0.05. Unless specified otherwise, parametric data are reported as mean and standard deviation (mean ± SD) and non-parametric data are reported as median and range. 

Data from each of the 17 horses were analysed for a correlation between VT_EIT_ and VT_SPIRO_ using simple linear regression for measurement M_10_, M_12_ and M_15_. The coefficients for slope and intercept were then used to estimate VT_EIT_ for subsequent data points M_A_, M_B_ and M_C_ from the measured ∆Z_breath_. The estimated VT_EIT_ and measured VT_SPIRO_ for M_A_, M_B_ and M_C_ were evaluated for agreement using Bland Altman analysis for repeated measurements (MedCalc Software Ltd. Ostend, Belgium). The mean difference (bias), 95% confidence interval of the bias (limits of agreement; LOA) and the 95% confidence limits of the LOA were calculated. 

To determine components of variance in the ∆Z_breath_ and VT_SPIRO_ correlation, the variables: leak volume; PIP; P_ET_CO_2_; VD_aw_/VT_SPIRO_; VT_alv_/VT_SPIRO_; CoV_VD;_ and DSS_S_; were explored using multiple regression with estimation of variance based on assessment of the coefficient of determination (R^2^) or adjusted R^2^ for models of two or more variables. Data for 12 horses was complete for all variables and was included in this multiple regression analysis.

All statistical analysis, unless specified was performed using SAS V 9.4.

## 3. Results

Of the 26 horses enrolled in the study, 17 data sets were included in data analysis (Figure 2). Data from nine of the 26 horses were not available: the EIT belt was positioned incorrectly (*n* = 2), the belt did not fit around the thorax of a horse (*n* = 1), EIT laptop failure which precluded data recording (*n* = 1), and finally, incomplete data due to short surgical procedure (*n* = 5).

Demographic data of the 17 horses included in final analysis are summarised in Appendix A. Mean body weight of the horses was 485 kg ± 51 kg with a mean chest circumference of 182 cm ± 11 cm and BCS ranging from 4/9 (*n* = 3) to 5/9 (*n* = 11) to 6/9 (*n* = 3). The median age was 39 (range 23–227) months.

Procedures performed in the 17 horses included arthroscopy (*n* = 7), castration (*n* = 8), sequestrum removal (*n* = 1), and stifle cyst removal (*n* = 1). The surgical and anaesthetic procedures were uneventful in all horses. Mean duration of anaesthesia was 107 ± 29 min. Of the 17 horses, 12 required dobutamine support to maintain mean arterial pressure between 70 and 80 mmHg (0.6 ± 0.2 µg/kg/min).

None of the breaths analysed had a leak that exceeded the 10% cut-off. 

### 3.1. Linear Relationship between VT_SPIRO_ and ∆Z_breath_

A positive linear correlation between ∆Z_breath_ and VT_SPIRO_ measured at M_10_, M_12_ and M_15_ was observed in each horse during progressive increases in delivered volume, with a mean correlation coefficient of 0.99 (range 0.90–0.99). The estimated intercepts and coefficients describing the correlation in each horse varied and did not support pooling the data and determining a global correlation between ∆Z_breath_ and VT_SPIRO_ (Figure 3; Appendix A). Our first hypothesis was then disproved. 

Descriptive statistics of ∆Z_breath_, VT_SPIRO,_ leak data, PIP, P_ET_CO_2,_ VD_aw_/VT_SPIRO_, VT_alv_/VT_SPIRO,_ CoV_VD_, DSS_S_ and F-Shunt are summarised in Table 1. Multiple regression of the 12 complete data set showed that the single variable that explained the most variance in the ∆Z_breath_ and VT_SPIRO_ relationship was VD_aw_/VT_SPIRO_ (43%). 

### 3.2. Estimation of VT_EIT_ from ∆Z_breath_

The tidal volume delivered ranged from 6.0 to 8.1 L between horses. Tidal volume was kept stable in 11 horses for measurement points M_A_, M_B_ and M_C_. Two horses required a decrease in VT and 4 required an increase to maintained P_ET_CO_2_ within the predefined range of 50 ± 5 mmHg. All changes made to VT for this purpose did not exceed 1 L. 

For each horse, VT_EIT_ was estimated from measured ∆Z_breath_ using the line of best fit (Appendix A). Bland–Altman analysis of the estimated VT_EIT_ and VT_SPIRO_ revealed a bias of 0.26 L with limits of agreement (95% confidence interval of LOA) of −0.36 (−0.69–0.155) to 0.88 (0.68–1.21) (Figure 4). The negative bias reflects a tendency to underestimate VT_EIT_ compared to VT_SPIRO_. 

## 4. Discussion

This study confirmed, in equine clinical cases, the presence of a strong positive linear correlation between tidal volume, measured with spirometry, and impedance changes, measured with EIT within each horse. This finding suggests that EIT impedance changes can be used clinically to monitor changes in tidal volume. However, a high variability of this relationship was observed between horses and, for this reason, it was not possible to pool the data and to determine a global correlation between the two variables.

Using the assessed individual correlation, this study reveals good agreement between the tidal volume estimated from EIT and tidal volume measured by spirometry. In these horses, the LOA were within 20% of the tidal volume measured with spirometry, supporting the potential for EIT to estimate tidal volume with sufficient accuracy for clinical application after evaluation of the individual linear relationship. 

### 4.1. Linear Correlation between VT_SPIRO_ and ∆Z_breath_


The first aim of the study was to explore the relationship between tidal volume, measured with spirometry, and impedance changes, measured with EIT, during clinical cases. A strong positive correlation between the two variables was previously determined experimentally in anaesthetised and mechanically ventilated horses with the authors reporting minimal variation in the relationship between horses [20]. In the present study, EIT was applied in equine clinical cases undergoing elective procedures. We documented a similar strong linear correlation between spirometry and EIT tidal volume. In contrast to the previous study however, we observed substantial variation in the responses of each individual horse to changes in the delivered volume with varying intercepts and slope of the regression line (Figure 3) Therefore, we were unable to define a single, global correlation for all horses in our study as hypothesised.

There are a number of possible explanations that may clarify the individual variability observed. Variables such as breed, age and body condition, could affect pulmonary mechanics, their response to a delivered volume and, therefore, be partly responsible for the individual variability of the relationship [36,37,38]. In this study, we enrolled horses of different breeds and age. Furthermore, a range of body sizes was supported by the large standard deviation in chest circumference in the population under exam (SD ± 11 cm).

The experimental settings between the experimental study and this clinical study also varied. In the experimental study no surgery was performed, delivered tidal volume ranged from 4 L to 10 L, higher peak inspiratory pressures were reached, and each volume was maintained for only 2 min [20]. In contrast to this, we used volumes and peak pressure that are used in clinical settings. A longer stabilisation period was allowed between measurements to permit adaptation of the lung tissue to changes in respiratory mechanics due to alterations in tidal volume [39]. This may add to the higher individual variability seen in our study as more time was allowed for alveolar dynamics to reach a steady state which has been shown to be different between individuals [40]. The stable lung conditions throughout our study period can be confirmed by stable values for centre of ventilation, dependent silent space and F-shunt over all measurement points.

The linear correlation observed in this study and previously described experimentally, suggests that changes in impedance are directly related to volume. The clinical relevance of this finding is that an increase in impedance corresponds to an increase in volume and vice versa, which gives the clinician the possibility to observe changes in tidal volume over time by assessing changes in impedance.

The effects of several variables of pulmonary function on the variance of tidal volume measured with EIT were explored. Preliminary examination of spirometry and EIT data from the first 5 horses suggested the presence of individual variability. This prompted investigation and recording of possible factors that could contribute to this variation. For this reason, volumetric capnography data was subsequently recorded in the remaining 12 horses. Of all the variables explored, airway dead space had the greatest influence on the variance of the relationship under examination. Airway dead space represents the volume of gas not taking part in gas exchange and being exhaled from the conducting airway [21,22,27]. This finding was expected as airway dead space represents the gas volume between the two measurement sites of the spirometer and EIT (Figure 1). This is even more true as the volume of the airway dead space is not constant, and it varies between individuals and between breaths [41]. Particularly relevant to this study are factors that may have affected airway dead space including the size of the patient, their age and the position of the neck. Airway dead space is also affected by flow characteristics [41]. According to Poiseuille’s equation, airway dimension (radius or diameter of a tube) affects both flow characteristics and resistance to flow. Endotracheal tube sizes were different between horses (Appendix A). Changes in diameter from the endotracheal tube to conducting airway may have changed the characteristics of the flow and altered the tidal volume relationship. 

None of the other recorded factors explored to determine components of variance in the tidal volume relationship were found to have influence on the linearity. As leaks in the system influence the accuracy of spirometric measurements, the flow-volume curves and the difference between inspiratory to expiratory volume were checked and inspiratory and expiratory tidal volume recorded, to be able to correct leaks by inflating the ETT cuff if necessary, prior to each data collection period. The leak was below 10% of the measured tidal volume in all measurements.

### 4.2. Estimation of Tidal Volume from EIT Measurements 

To compensate for the individual variation between horses, the line of best fit was used to make a clinically acceptable estimate of tidal volume in litres from EIT impedance changes measurements. The limits of agreement between estimated tidal volume from EIT and tidal volume from spirometry were within 20% of the tidal volume measured using spirometry. Currently, in veterinary medicine, there are no established accuracy standards for clinical applicability of multiple methods for monitoring vital parameters. Instead, we are reliant on human accuracy standards. In human studies, an agreement of ±15% between measurement methods for lung volume has been considered acceptable for clinical application [23,24]. The measured agreement of less than 20% in this study is encouraging and supports the potential of EIT to estimate VT in litres in anaesthetised and mechanically ventilated horses after the individual correlation has been established. 

We observed a tendency for the estimated tidal volume from EIT to be lower than tidal volume measured with spirometry. This is expected, as spirometric measurements are taken proximally at the level of the mouth and includes airway dead space as explained above and shown in Figure 1. EIT, in contrast, represents cross-section of the lungs at the level of the 5th–6th intercostal space and does not include airway dead space. In a cattle study, which used EIT measurement to evaluate the effect of PEEP on tidal volume, it was observed that airway dead space contributed to the explanation of variance together with peak inspiratory pressure up to an adjusted R^2^ of 0.76 [42]. 

Future study should explore the relationship between alveolar ventilation and tidal volume measured with EIT. According to our observations of airway dead space, it would not be surprising if impedance changes would have a better agreement with alveolar ventilation, than with tidal volume itself.

### 4.3. Clinical Observation of Ease of Use of EIT

To the authors knowledge, this is the first study on EIT in clinical equine cases. The electrode belt was easily placed around the thorax of sedated horses after gel was applied. A soft elastic bandage was placed around the belt to prevent its movement during induction and hoisting of the horse. The belt was visually inspected before positioning the horse on the surgical table. Overall, the electrode-skin contact was good in all horses without prior hair clipping. 

Belt mispositioning, due to operator error and rough induction, caused the exclusion of two horses from our study [43]. Our experience confirmed that accurate EIT belt positioning was a key factor to measurement consistency. EIT laptop failure prevented data recording in one horse. This was resolved by updating the software. 

In horses that underwent castration (*n* = 8), EIT measurement artefacts were observed when electrocautery was used making data collection impossible at the same time. This resulted in less than 10 analysed breaths in three horses with one horse having only 3 breaths analysed at one measurement point. The use of electrocautery needs to be considered when using EIT as monitoring tool in a theatre setting. Prompt and consistent communication with the surgeon was implemented to limit electrical artifacts during data collection.

### 4.4. Limitations of the Study

The linearity of the relationship and estimation of tidal volume rely on the accuracy of the spirometric measurement adopted. According to the manufacturer specifications, the flow sensor used has an accuracy of ±3%. A recent in vitro study highlighted an increase in the margin of errors in calibrated Pedilite flow sensors following increasing volumes delivered [44]. This observation is consistent with what has been reported in studies related to the accuracy of FPD in association with the spirometer used in this study [8,10].

We tested the accuracy of the spirometer with a calibration syringe before each anaesthesia. The margin of error identified was in agreement with manufacturer specifications. In general, accuracy of spirometry is affected by gas viscosity, temperature and the presence of water vapour [45]. These variables change when considered in in vivo and in vitro settings and may affect volume measurements. When performing accuracy tests on measurement methods, it should be noted that some variables can be controlled in experimental settings, but not in clinical conditions, which means that the accuracy of any measurement methods may vary depending on the conditions implemented.

Another limitation of the study is attributed to the use of the FPD. The FPD enabled the application of standard human spirometry to large animal anaesthesia in a simple and reproducible way. Previous research supported the use of FPD paired with a commercially available spirometer as an accurate and reliable way to measure tidal volume in anaesthetised horses, therefore this method was chosen for this study as the reference method [8,10]. The downside of using an FDP is that a correction factor of 4 is required for volume measurements [10]. Applying the correction factor also causes the measured error to be multiplied by 4.

The differences in demographic data together may explain the individual variability in the relationship we observed. Individual features such as breed, sex, age, weight and chest circumference, BCS were measured but not included in the statistical analysis. Three consecutive measurement points were used to estimate tidal volume from EIT; however, we did not evaluate time as an influencing variable. Future studies should include more data point to further characterised the relationship and also explore the influence of time in the stability of the relationship.

A final limitation was the differences in the number of selected breaths analysed at each measurement point for different horses. Electrical interferences caused by the use of the electrocautery, affected EIT measurements and impaired their interpretation, making it necessary to exclude these breaths. This should be taken into account in future studies where EIT is used in clinical cases.

## 5. Conclusions

This study demonstrates in clinical equine patients a strong positive linear correlation between impedance changes measured with EIT and tidal volume measured with spirometry. According to our findings, EIT impedance changes can also be used to detect changes in tidal volume. However, there was marked variability in this relationship between individual horses. When the individual line of best fit from each horse was used to estimate VT, the agreement between estimated VT_EIT_ and measured VT_SPIRO_ was within LOA of 20%. This suggests that EIT has the ability to estimate VT in anaesthetised and mechanically ventilated horses.

## Figures and Tables

**Figure 1 animals-11-01350-f001:**
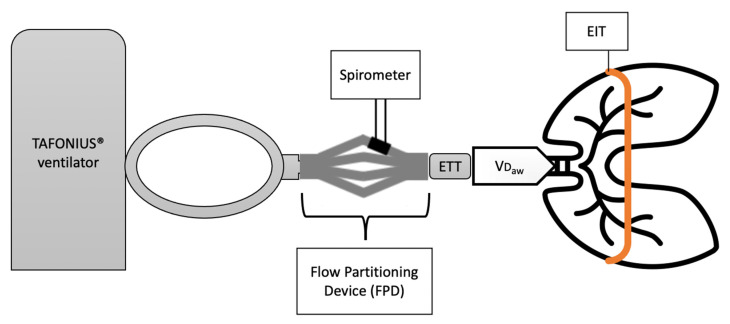
Schematic diagram of the equipment configuration adopted for data collection. From the left to the right: large animal ventilator (Tafonius Junior, Vetronics, UK), and its circle system are connected to the endotracheal tube (ETT) of the patient. In between the Y-piece of the circle system and the ETT is represented the Flow Partitioning Device (FDP) which enabled VT measurements at the level of the incisors. An electrode belt, placed around the thorax, allows for ∆Z measurements during the breathing cycle. Airway dead space (VD_aw_) represents the volume of gas exhaled from the conducting airway that does not participate to gas exchange. Spirometric and EIT measurements are taken from two different levels (mouth and thorax, respectively) highlighting why changes in impedance do not account for VD_aw_ as much as spirometry does.

**Figure 2 animals-11-01350-f002:**
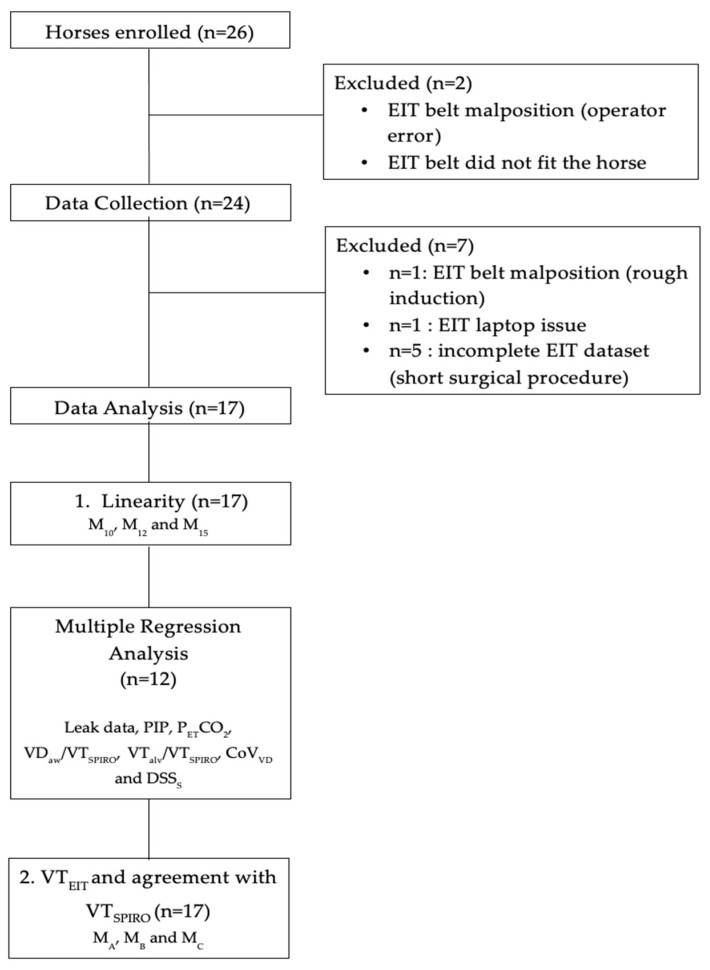
Flow chart showing how many horses were enrolled and how many were excluded from data collection and analysis.

**Figure 3 animals-11-01350-f003:**
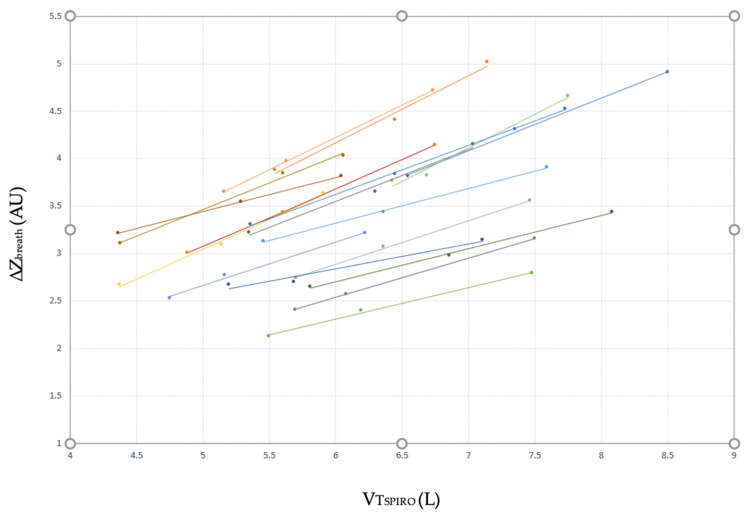
Simple linear regression plots between impedance measured using EIT (∆Z_breath_) and tidal volume measured using spirometry (VT_SPIRO_) at measurement M_10_, M_12_, and M_15_ in 17 horses for delivered tidal volumes of 10, 12 and 15 mL/kg.

**Figure 4 animals-11-01350-f004:**
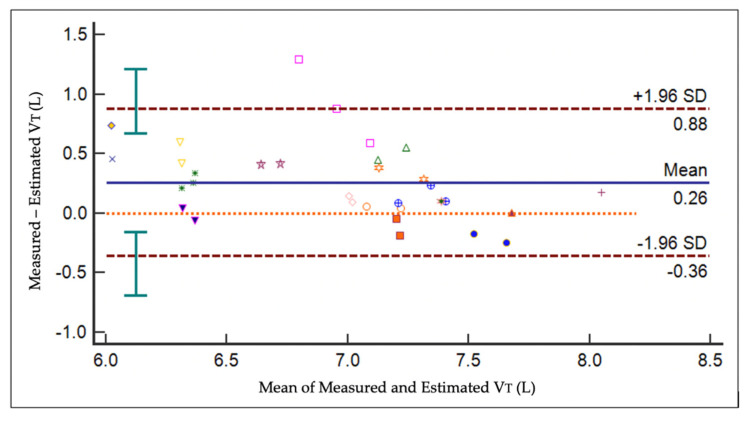
Bland Altman plot of the difference in measured versus estimated VT against the mean of the measured and estimated VT in 17 horses. The solid line represents the mean difference (bias) and brown dotted lines represent limits of agreement (LOA). The 95% confidence intervals of the LOA are shown by green bars.

**Table 1 animals-11-01350-t001:** Mean ± standard deviation of parameters measured with spirometry, volumetric capnography and EIT for measurement point M_10_, M_12_, M_15_, M_A_, M_B_ and M_C_ in 17 horses. VD_aw_/VT_SPIRO_ and VT_alv_/VT_SPIRO_ were derived from *n* = 12 horses. F-shunt for measurement point M_12_ and M_B_ is also reported.

Variables	M_10_	M_12_	M_15_	M_A_	M_B_	M_C_
∆Z_breath_ (AU)	3.0 ± 0.5	3.4 ± 0.6	3.9 ± 0.7	3.8 ± 0.6	3.8 ± 0.5	3.9 ± 0.9
VT_SPIRO_ (L)	5.3 ± 0.7	6.0 ± 0.6	7.1 ± 0.7	7.1 ± 0.5	7.1 ± 0.4	7.0 ± 0.4
Leak Data (mL)	39 ± 28	31 ± 30 *	54 ± 47 *	39 ± 36 *	46 ± 37 *	68 ± 52
PIP (cmH_2_O)	16 ± 4 *	19 ± 3.0	22 ± 4 *	23 ± 4	24 ± 5 *	24 ± 5
P_ET_CO_2_ (mmHg)	57 ± 8	52 ± 10	48 ± 10	47 ± 8	45 ± 9	43 ± 6
VD_aw_/VT_SPIRO_	0.6 ± 0.04	0.58 ± 0.06	0.52 ± 0.05	0.51 ± 0.04	0.51 ± 0.05 *	0.50 ± 0.04
VT_alv_/VT_SPIRO_	0.40 ± 0.05	0.43 ± 0.05	0.44 ± 0.13 *	0.48 ± 0.04	0.48 ± 0.05	0.50 ± 0.04
CoV_VD_ (%)	41 ± 3.3	40 ± 2.7	41 ± 2.5	41 ± 2.8	42 ± 2.8	41 ± 2.8*
DDS_S_ (%)	14 ± 12	14 ± 11	14 ± 12	12 ± 10	15 ± 13 *	11 ± 9 *
F-SHUNT (%)	*n*/a	22 ± 0.06	*n*/a	21 ± 0.06	*n*/a	*n*/a

∆Z_breath_, impedance changes measured with EIT; VT_SPIRO_, tidal volume measured with spirometry; PIP, peak inspiratory airway pressure; P_ET_CO_2_, end-tidal carbon dioxide tension; VDaw/V_TSPIRO_, airway dead space over tidal volume; VT_ALV_/V_TSPIRO_ alveolar ventilation over tidal volume; CoV_VD_, ventro-to-dorsal centre of ventilation; DDSs, dependent silent spaces; * *p* ≤ 0.05 the test rejects the hypothesis of normality.

## Data Availability

Data is contained within the article and Appendix A provided.

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
