# Peer review of "Use of Electrical Impedance Tomography (EIT) to Estimate Tidal Volume in Anaesthetized Horses Undergoing Elective Surgery"

_animals, 2021, doi:10.3390/ani11051350_

Round 1
Reviewer 2 Report
Dear Authors
thank you for submitting this paper. This new version is clearer and more undestandable than the previous one, I could find only a type error at line 100. No further comments
Best wishes
This manuscript is a resubmission of an earlier submission. The following is a list of the peer review reports and author responses from that submission.
Round 1
Reviewer 1 Report
MS Review
Title: Use of electrical impedance tomography (EIT) to estimate tidal volume in anaesthetized horses undergoing elective surgery
General comments:
Measuring Vt with EIT is of questionable benefit since it must be calibrated. This manuscript is repeating the results of a previous experimental study and I am not sure whether it is sufficiently original or whether it adds to our current knowledge enough to justify publication. What is the point in repeating a study using a different cohort of horses? Was the concept different in any way? The study is lacking proper justification.
The manuscript is poorly written and uses imprecise terminology and arguments (please see my specific comments). Many tables and figures contain repeated and redundant data that has nothing to do with the study aim and represent no learning point. Showing data of individual animals is not appropriate.
Specific comments:
L21: “volume of air in litres moved in and out the lungs with each breath”
Are you sure this definition is generally acceptable as it stands? The definition should not be limited to air but include other gases that horses may inhale under anesthesia. Would you like to limit Vt to the volume moving in/out of the lungs or would you include the airways too?
L23: please be more specific: “conventional tidal volume monitors”
L24: correlation rather than relationship
L25: What do you mean by “known tidal volume”?
L27: individual calibration is more appropriate. One shall not assume that such relationship will always be linear despite it was in the current study.
L29: usefulness instead of use
L36: I suggest using slope instead of coefficient.
L37: please define: measurements (MA, MB and MC)
L41: In this study you measured Vt and not Vt changes. Again: individual calibration is better…
L47: Very arguable statement and the referenced Nyman paper doesn’t prove that Vt measurement is improving ventilation. Do you mean that ventilation of anesthetized horses will be adequate if Vt is accurately monitored? How do you define ventilation? Would you include V/Q matching or would you limit it to PaCO2 values being in the normal range. The latter is easy to ensure without Vt monitoring.
L49 What kind of spirometry?
L51: A large animal spirometry sensor that is compatible with human spirometry monitors is possible to obtain from less than 1000 USD. It is neither complex nor expensive. On the contrary, EIT is much more complex and expensive.
L68: Based on your results, EIT is not an alternative to Vt measurement since you need spirometry anyway to calibrate EIT.
L70-75: I disagree with this argument. Spirometry measures the volume flowing through the sensor regardless of where it is coming from. On the other hand, impedance measurements reflects the tidal expansion of the lungs, however, it may be mostly attributable to the expansion of the bronchial system and not the alveoli that hardly changes volume during tidal breathing. And since you proved that spirometry and EIT Vt are strongly correlated therefore they should refer to the same thing as long as this relationship is maintained.
L81: Is this study sufficiently original to deserve publication? Why did the authors repeat that experimental study? Citation is needed for that experimental study.
L97: I doubt that lung health can be assumed without a BAL.
L179: What do volumetric capnography and blood gas measurements have to do with the study aims?
L194: delta Z is defined for each pixel right? Do you mean you used the sum or the average of all pixels at each time point?
L194: please spell out ventro-dorsal
L199: the point around which ventilation occurs? It cannot be “around” since it is a 2D concept therefore it could only be before and after right? And it is not “only around” but also right there. I just don’t really understand what the Authors are trying to say.
L200: What kind of EIT image are the Authors referring to? Each pixel as an SD of delta Z over time?
L202: VD extension of lung region? I suggest the Authors should describe what they did by terms that would be understood by a reader not familiar with EIT analysis.
Why did the Authors calculate VD COV despite it’s not related to the study aim? Why didn’t they calculate LV COV too?
L206: Why was it necessary to average each breath rather than using SD images over the whole duration of sampled breaths?
L207: How do you know those were lung regions?
L210: How would a collapsed lung be ventilated and change delta Z with each breath? Are you sure the cyclic recruitment occurred? Doubtful when using clinically relevant PIP values.
L215: If the readers know this technology, they don’t need this information since it is a minor mathematical detail. If they don’t know it they won’t understand what the Authors mean by “accounting for the split”. I would consider them Vt measurements after this multiplication.
L236: The LOA is NOT the same as the 95% confidence interval of the bias!
L240: what is “leak data”? Do you mean leak volume?
L276: “All changes were made within 1 L increment or decrement of VT.” What does it mean?
L280: Where does this 17% discrepancy coming from. Please describe more precisely.
Table 1: it is not necessary to show the slope and intercept for each single horse. A summary statistics of these coefficients would suffice.
Figure 3: There is no learning point from this figure since it shows the same data as table 1, therefore it is redundant. Knowing the range of R2 values strong correlation is obvious, no need to see the calibration curves of each individual horse.
Table 2: Most of this table shows data that are irrelevant for the study aims and hypothesis testing. Therefore represent no learning point. Please focus on your main findings and try not inflating data. * Means data was not normally distributed: why did you express it as mean/SD?
Table 3: redundant data that is the same as also shown on Figure 4. Please delete this table.
L306: Why are the CI95 bars not symmetrical above and below the LOA?
L317: How do you justify clinical application if you need the spirometry anyway to calibrate EIT?
L466” It cannot become a surrogate if it is bound to calibration.
Author Response
Dear Reviewer,
Thank you for taking the time to review so thoroughly the manuscript.
We appreciated all suggestion made to improve the scientific quality of the manuscript. We have attempted to address all the comments and concerns with our response in bold below. We also marked all changes in the manuscript following the editor’s guideline.
Again, thank you for your contribution.
the Authors
Reviewer General comments:
Measuring Vt with EIT is of questionable benefit since it must be calibrated. This manuscript is repeating the results of a previous experimental study and I am not sure whether it is sufficiently original or whether it adds to our current knowledge enough to justify publication.
- What is the point in repeating a study using a different cohort of horses?
- Was the concept different in any way?
- The study is lacking proper justification.
We want to thank the Reviewer for making us aware that these important points were not clear.
This paper aimed to test, clinically, the performance of EIT in anaesthetised horses.
Previously the relationship between ∆Z measured using EIT and tidal volume measured using spirometry was tested in experimental settings under standardised and controlled conditions. As the previous study was performed in horses that were euthanised at the end of anaesthesia we were able to assess the relationship over a wide range of tidal volume that would not be feasible in clinical cases. The previous study also used a shorter 2-minute stabilisation period between tidal volume changes.
Thus, we wanted to determine if the same relationship would exist during application of clinically relevant tidal volume and when a longer stabilisation period was used. This was also the first study in which VT was estimated by the EIT from ∆Z measurements once the relationship was calculated.
We added a sentence in the conclusion of the introduction in the attempt to make this clearer.
The manuscript is poorly written and uses imprecise terminology and arguments (please see my specific comments).
We have attempted to address all the Reviewer specific comments below.
Many tables and figures contain repeated and redundant data that has nothing to do with the study aim and represent no learning point.
We tried to delete data and avoid double reporting and point out our outcome measurements for the reader.
Showing data of individual animals is not appropriate.
We deleted all individual data from the revised manuscript.
Specific comments:
- Simple Summary
L21: “volume of air in litres moved in and out the lungs with each breath”
Are you sure this definition is generally acceptable as it stands? The definition should not be limited to air but include other gases that horses may inhale under anesthesia. Would you like to limit Vt to the volume moving in/out of the lungs or would you include the airways too?
We understand this being an oversimplification, but the guidelines for simple summary invite authors to “describe their work simply and concisely” addressing it “for a lay audience”. For the purpose of precision, we amended it as follow: “volume of gas in litres moved in and out the airways and lungs with each breath”.
L23: please be more specific: “conventional tidal volume monitors”
We rephrased the sentence as follow “…there was a positive linear relationship between tidal volume measurements obtained with spirometry and impedance changes measured by EIT”
L24: correlation rather than relationship
We note the Reviewer's suggestion, however, the Authors would prefer to call it relationship when it is referring to the association of the two variables in broader terms and correlation when it is referring to the statistical methodology.
L25: What do you mean by “known tidal volume”?
Thank you for highlighting the lack of clarity in this statement. We have reworded it to “with a measured tidal volume”.
L27: individual calibration is more appropriate. One shall not assume that such relationship will always be linear despite it was in the current study.
We agree with the Reviewer that the linearity should not be taken for granted. For this reason, we removed the expressions “linear” and left “individual relationship”. However, we never calibrated the EIT.
- Abstract
L29: usefulness instead of use [L30]
Amended [L30]
L36: I suggest using slope instead of coefficient. [L38]
Amended
L37: please define: measurements (MA, MB and MC)
We want to thank the Reviewer for making us aware that this important point was not clear.
We have reworded both the abstract and data collection sections. We hope this addressed the Reviewer’s concern.
To answer to the Reviewer: MA, MB, MC, are measurement points collected in Part 2 of the study. The Reviewer is directed to the data collection paragraph.
L41: In this study you measured Vt and not Vt changes.
This has been reworded. [L41]
Again: individual calibration is better…
The Reviewer is directed to our answer to comment L27.
- Introduction
We rewrote the whole introduction based on the Reviewer’s comments.
We still include below the answers to the Reviewer’s specific comments on our original version.
We hope this addressed the Reviewer’s concern.
L47: Very arguable statement and the referenced Nyman paper doesn’t prove that Vt measurement is improving ventilation.
We agree that this reference is not appropriate for the statement made and have changed the reference accordingly.
Do you mean that ventilation of anesthetized horses will be adequate if Vt is accurately monitored?
We don’t think it will be necessarily adequate, but it will be an important additional monitoring variable.
How do you define ventilation?
Ventilation is defined as the process through which a volume of gas is transported to and from the lungs to provide oxygen and eliminate carbon dioxide. Not all tidal ventilation takes part in gas exchange: just alveolar ventilation does.
We added a sentence to clarify this in the introduction.
Would you include V/Q matching or would you limit it to PaCO2 values being in the normal range. The latter is easy to ensure without Vt monitoring. [L48]
We would define it as PaCO2 value within the physiologic range. However, PaCO2 measurements via blood gas analysis are intermittently assessed. PETCO2, on the other side, is not a reliable monitoring of ventilation adequacy in anaesthetised horses because of a broaden partial pressure gradient between PETCO2 and PaCO2.
Spirometry and EIT would provide continuous monitoring of tidal ventilation. We added this to the new introduction
L49: What kind of spirometry?
The paragraph has been rephrased to clarify [L80]
L51: A large animal spirometry sensor that is compatible with human spirometry monitors is possible to obtain from less than 1000 USD. It is neither complex nor expensive. On the contrary, EIT is much more complex and expensive.
We have reworded this section to clarify our desire to investigate the EIT technology.
We also added some more considerations based on other Reviewers' comments and suggestion to the rewritten introduction. [L121]
L68: Based on your results, EIT is not an alternative to Vt measurement since you need spirometry anyway to calibrate EIT.
We rephrased the sentence to make clear that this is what it was suggested in previous research. This study is meant to investigate exactly this. [L107]
L70-75: I disagree with this argument. Spirometry measures the volume flowing through the sensor regardless of where it is coming from. On the other hand, impedance measurements reflect the tidal expansion of the lungs, however, it may be mostly attributable to the expansion of the bronchial system and not the alveoli that hardly changes volume during tidal breathing. And since you proved that spirometry and EIT Vt are strongly correlated therefore they should refer to the same thing as long as this relationship is maintained. [L110-115]
We have attempted to clarify our statement and the reader is directed to Figure 1 illustrating the set-up.
We agree with the Reviewer’s statement regarding spirometry but are not sure if we can fully agree with the EIT statement. It has been shown that the impedance change measured by EIT comprises gas flowing through the airways plus gas expanding the lung tissue (alveoli) during each breath.
- Frerichs I, Amato MBP, van Kaam AH, Tingay DG, Zhao Z, Grychtol B, et al. Chest electrical impedance tomography examination, data analysis, terminology, clinical use and recommendations: consensus statement of the TRanslational EIT developmeNt stuDy group. Thorax. 2016)
- Costa EL, Borges JB, Melo A, Suarez-Sipmann F, Toufen C Jr, Bohm SH, Amato MB. Bedside estimation of recruitable alveolar collapse and hyperdistension by electrical impedance tomography. Intensive Care Med. 2009.
- Moerer, Onnen; Hahn, Günter; Quintel, Michael Lung impedance measurements to monitor alveolar ventilation, Current Opinion in Critical Care: June 2011 - Volume 17 - Issue 3 - p 260-267
L81: Is this study sufficiently original to deserve publication? Why did the authors repeat that experimental study? Citation is needed for that experimental study.
The novel insights of the study have been highlighted at the end of the introduction after aims and hypothesis. [L138]
- Materials and Methods
L97: I doubt that lung health can be assumed without a BAL
We have rephrased the statement. [L195-199]
L179: What do volumetric capnography and blood gas measurements have to do with the study aims? [L299]
We hope that with the rephrasing of the introduction it becomes clear why we included volumetric capnography as measurement option for VDaw and VTalv in our study. Blood gas measurements were used to calculate f-shunt which was used to show the lung condition over the study period (L395-397)
L194: delta Z is defined for each pixel right? Do you mean you used the sum or the average of all pixels at each time point?
Thank you for making us aware of this oversight. We used the sum of delta Z of all pixels. We added the word ‘total’ to our explanation [L323]
L194: please spell out ventro-dorsal [330]
Amended
L199: the point around which ventilation occurs? It cannot be “around” since it is a 2D concept therefore it could only be before and after right? And it is not “only around” but also right there. I just don’t really understand what the Authors are trying to say.
Thank you to highlighting this, we acknowledge that this sentence was poorly worded. We rephrased it as follow:
L328: “Centre of Ventilation (CoV) represents the focal area of the functional EIT image where ventilation occurs and is a parameter used to quantify the overall distribution of ventilation”
L200: What kind of EIT image are the Authors referring to? Each pixel as an SD of delta Z over time? [328]
Thank you for making us aware of this oversight – we added the word ‘functional’ as given by the consensus statement of the TRanslational EIT developmeNt stuDy group (2016)
L202: VD extension of lung region? I suggest the Authors should describe what they did by terms that would be understood by a reader not familiar with EIT analysis.
We agree with the reviewer that this is unclear. We added the following statement in the revised version [L330]:
The ventro-dorsal CoV (CoVVD) is generally expressed as a percentage of the ventro-dorsal extension of the predefined region of interest (ROI) within the functional EIT image that represents the lung regions.
Why did the Authors calculate VD COV despite it’s not related to the study aim? Why didn’t they calculate LV COV too?
COVVD was calculated and analysed in the attempt to explain the individual variability of the relationship. COVLV was recorded but not included into the final statistical analysis because all our horses were in dorsal recumbency and we did not expect this variable to change significantly over our study period.
L206: Why was it necessary to average each breath rather than using SD images over the whole duration of sampled breaths? [337]
Ibex which was the software used for analysis of the EIT raw data only allows breath by breath analysis – so for this reason we needed to analyse each breath and then take the average.
L207: How do you know those were lung regions?
Thank you for making us aware of this – we reworded this sentence [L339]:
Silent spaces refer to pixels within the ROI with an impedance change < 10% of the maximal impedance change of any pixel within the ROI.
L210: How would a collapsed lung be ventilated and change delta Z with each breath? Are you sure the cyclic recruitment occurred? Doubtful when using clinically relevant PIP values.
We reworded this sentence [339-340]:
For the purpose of this study, only dependent silent spaces were calculated (DSSS), which indicates poorly ventilated (pixel in the ROI with 0 > ∆Z 10% of maximal ∆Z within ROI) or collapsed (pixel in the ROI with ∆Z = 0%) regions of the dependent lung. Like COVVD, DSSS were calculated as an average of all breaths selected, at each measurement point.
L215: If the readers know this technology, they don’t need this information since it is a minor mathematical detail. If they don’t know it they won’t understand what the Authors mean by “accounting for the split”. I would consider them Vt measurements after this multiplication.
In the instrumentation part, it is briefly explained how the flow partitioning device works. With that explanation in mind an attentive reader, with or without knowledge about the technology, should be able to understand what we meant by “accounting for the split”. We believe that “multiply by 4” is not a minor mathematical detail because if omitted might mislead the reader or lead to erroneous results future researchers. [353-354]
L236: The LOA is NOT the same as the 95% confidence interval of the bias! [377]
It is standard methodology and nomenclature that the LOA is defined by the confidence limits of the bias. In this case, we have defined them as the 95% CI – in some cases someone might decide on the 90% or the 99% CI.
L240: what is “leak data”? Do you mean leak volume?
This has been clarified in the text.
L276: “All changes were made within 1 L increment or decrement of VT.” What does it mean?
The sentence has been rephrased [418].
L280: Where does this 17% discrepancy coming from. Please describe more precisely.
We were trying to reflect the extent of the discrepancy but have deleted that sentence since we agree, it is confusing and redundant to the graphs and descriptions of the extent of the LOA's. [422]
- Table and Figures
Table 1: it is not necessary to show the slope and intercept for each single horse. A summary statistics of these coefficients would suffice.
The authors were using this table to show that each horse was different but agree it is redundant to the figure. We have moved Table 1 to supplementary materials as believe its inclusion does give transparency to the results.
Figure 3: There is no learning point from this figure since it shows the same data as table 1, therefore it is redundant. Knowing the range of R2 values strong correlation is obvious, no need to see the calibration curves of each individual horse.
The Authors included this figure to allow the reader to see the linearity in the relationship but also the variability in the slope across horses as this is the key finding of part 1. With the deletion of Table 1, we would prefer to keep the figure in the main body of the manuscript.
Table 2: Most of this table shows data that are irrelevant for the study aims and hypothesis testing. Therefore, represent no learning point. Please focus on your main findings and try not inflating data.
This table was included to summarise the variables that were included in the multiple regression analysis. It may be preferable to also add this table to the supplementary materials but its inclusion in this way again allows transparency of the results.
* Means data was not normally distributed: why did you express it as mean/SD?
There were only a select few data subsets that were not normally distributed and, bearing in mind that the mean and SD are a reasonable, robust description of central tendency even in the face of non-normality, we chose to report it similarly for consistency. This data may, in fact, be removed from the main manuscript and contained within the supplementary materials depending on editing decisions so perhaps this is less important in that context also. If the Editor would prefer, we could change them to median and range.
Table 3: redundant data that is the same as also shown on Figure 4. Please delete this table.
We deleted table 3 following the Reviewer’s suggestion and added it to the supplementary materials.
Figure 4: Why are the CI95 bars not symmetrical above and below the LOA?
MedCalc uses the delta method to recover the variance estimates needed for the confidence intervals of the LOA. This creates a right-shifted confidence limit for the lower LOA and a left-shifted confidence limit for the upper LOA. Thus, they appear asymmetric on the graph.
Zou GY (2013) Confidence interval estimation for the Bland-Altman limits of agreement with multiple observations per individual. Statistics in Medicine 22:630-642.
- Discussion
L317: How do you justify clinical application if you need the spirometry anyway to calibrate EIT? [L456]
Part 1 of our study shows that every change in impedance represents a change in the same direction of the measured tidal volume. This allows the clinical diagnosis of a decrease or increase in tidal volume over time by measuring impedance change (L??? in discussion)
The linear relationship in a patient can also be established by the delivery of two or more known volumes which would make a spirometer unnecessary. However, the authors are reluctant to make this statement in this paper as we have insufficient evidence to underline this so far.
L466” It cannot become a surrogate if it is bound to calibration. [L610]
We deleted this sentence in the revised manuscript
Reviewer 2 Report
The study is well designed the objective and the hypothesis are clearly stated, the measuring techniques are valid and reliable according to the references.
The abstract is concise and accurately summarizes the essential information contained within the manuscript.
The Introduction provides a precise scientific context as well as an explanation of rationale.
Electrical impedance tomography (EIT) is a radiation-free functional imaging modality that enables non-invasive monitoring of regional pulmonary ventilation and indirectly perfusion; It would be useful to state which are the advantages of using EIT against spirometry in equine species, the explanation provided by the author between line 50-52 is not complete, since EIT is also very expensive.
Material and Methods: On Data Collection section, the description of the first part of the study is well explained, but on the second part, it is unclear how the measurement MA, MB, MC were taken in relation to the surgical or the anaesthetic procedure. Also, may the author explain why on table 3 for each horse those measurements are different (eg: horse 1 recorded measurements point MA, Mc and horse 5 recorded measurements point MA, MB, Mc)?
Line 19: Please remove stable and representative, making the sentence smoother to read
Line 229: Please change the symbol with (mean ± SD)
Author Response
Dear Reviewer,
Thank you for taking the time to review the manuscript.
We appreciated the positive feedback and the suggestion made to improve the scientific quality of the manuscript.
We have attempted to address all the comments and concerns with our response in bold below. We also marked all changes in the manuscript following editor’s guideline.
Again, thank you for your contribution.
the Authors
Electrical impedance tomography (EIT) is a radiation-free functional imaging modality that enables non-invasive monitoring of regional pulmonary ventilation and indirectly perfusion; It would be useful to state which are the advantages of using EIT against spirometry in equine species, the explanation provided by the author between line 50-52 is not complete, since EIT is also very expensive.
We agree with the Reviewer about EIT being an expensive technology. We have reworded the first section of the introduction based on similar comments from another Reviewer – we added these considerations and hope that we have addressed the concerns of the Reviewer.
Material and Methods: On Data Collection section, the description of the first part of the study is well explained, but on the second part, it is unclear how the measurement MA, MB, MC were taken in relation to the surgical or the anaesthetic procedure.
We want to thank the Reviewer for making us aware that this important point was not clear. We have rewarded both the abstract and data collection sections to clarify that MA, MB, MC were measurement points.
Also, may the author explain why on table 3 for each horse those measurements are different (eg: horse 1 recorded measurements point MA, Mc and horse 5 recorded measurements point MA, MB, Mc)?
We removed Table 3 at the request of one of the other Reviewers.
In answer to the Reviewer’s question: if we were unable to select at least 3 consecutive and artefact fee breaths during the data collection period we excluded this time point from further analysis (e.g. MB in horse 1).
Also, if the procedure was long enough, we were able to collect all measured points MA, MB and MC. However, due to some short procedure times we were unable to have 3 data collection periods. This is the “drawback” of a clinical study.
Line 19X: Please remove ‘stable and representative’, making the sentence smoother to read [Data Extraction Section]:
We agree with the Reviewer regarding the sentence.
We have removed “representative” but kept “stable” as we believe it is important to emphasise that those breath were free of artifacts.
Line 229: Please change the symbol with (mean ± SD) [Statistical analysis Section]:
Amended
Reviewer 3 Report
Dear Authors,
thank you for submitting this well written and interesting paper.
I just have little considerations.
I didn't understand how you managed tidal volume changes. In line 174 you state that "The tidal volume was adjusted according to clinical assessment to maintain end tidal carbon dioxide tension (PETCO2) at 50 ± 5 mmHg. " Was indeed TV increased and mantained up to 15 min + 2 min for data collection? did you made any changes in the meanwhile? Where all horses maitaning the end tidal carbon dioxide tension between which limits? I think it is interesting to know because that could contribute to pulmonary vascular changes, V/Q mistmach and then distribution of gasses. Would you mind to clarify that to me please?
Furthermore did you notice if different premedication and maintenance agent had an influence in the results?
thank you in advance for your work
BW
Author Response
Dear Reviewer,
Thank you for taking the time to review the manuscript.
We have attempted to address all the comments and concerns with our response in bold below. We also marked all changes in the manuscript following the editor’s guideline.
Again, thank you for your contribution and positive feedback.
the Authors
I didn't understand how you managed tidal volume changes. In line 174 you state that "The tidal volume was adjusted according to clinical assessment to maintain end tidal carbon dioxide tension (PETCO2) at 50 ± 5 mmHg.
Was indeed TV increased and mantained up to 15 min + 2 min for data collection?
Yes – tidal volume was adjusted to maintain PETCO2 at 50 ± 5 mmHg for the entirety of part 2 of the study.
We tried to clarify this in the revised version of our manuscript.
did you made any changes in the meanwhile?
Not during part 1 of the study in which tidal volume was changed after each data collection period from 10 to 12 and 15 mL/kg (M10, M12 and M15).
During part 2 of the study tidal volume was left unchanged unless it was required to adjust PETCO2 levels to within the specified range.
Where all horses maitaning the end tidal carbon dioxide tension between which limits?
No, not all horses. The Reviewer is directed to the Results section at paragraph 3.2 - Estimation of VTEIT from ∆Zbreath:
“…Tidal volume was kept stable in 11 horses for measurement points MA, MB and MC. Two horses required a decrease in VT and 4 required an increase to maintained PETCO2 within the predefined range of 50 ± 5 mmHg. All changes made to VT for this purpose did not exceed 1L.”
I think it is interesting to know because that could contribute to pulmonary vascular changes, V/Q mistmach and then distribution of gasses. Would you mind to clarify that to me please?
The data collection section was rephrased to clarify this.
We want to thank you the Reviewer to make us aware that this important point was not clear.
Furthermore did you notice if different premedication and maintenance agent had an influence in the results?
The present study did not look objectively at the influence of different anaesthetic protocols on the outcome. From a subjective point of view, we did not have the impression that the protocol chosen affected the outcome of our study.
Reviewer 4 Report
Dear Authors,
mani thanks for your submission in this journal
I'm very sorry but i can't accept this study, although the subject is really interesting, I believe that the number and heterogeneity of the cases presented are not sufficient to adequately support the results.
I suggest you to increase the clinical cases on the basis of a preliminary power analysis to better evaluate the effect size
Author Response
Dear Reviewer,
Thank you for your brief comment on our study but it is difficult to understand exactly what you are referring to.
There was no effect size in the study that we are targeting; the study is descriptive, using the analysis of the cases to describe the relationship we identified between the EIT and the tidal volume. It is the heterogeneity of the horses that is a key finding and directs the interpretation of the result. To find the “on average” result over a large cohort of horses would really mask the heterogeneity and create a situation where over and under estimation would be very likely when applied to an individual horse. Accepting the heterogeneity and working with each horse as its own “control” is an effective way to manage this data.
We hope this help to address your concern.
Thank you again,
The Authors